



# On the transition from strong to weak constraint 4DVar using a simple one-dimensional advection equation for a passive tracer

Noureddine Semane[1]

[1]European Centre for Medium-Range Weather Forecasts, Shinfield Park, Reading, RG2 9AX, United Kingdom

**Correspondence:** Noureddine Semane (noureddine.semane@ecmwf.int)

**Abstract.** In contrast to strong constraint 4DVar, the weak constraint takes into account the model imperfection in the minimisation process. Relaying on a simple one-dimensional advection equation for a passive tracer, this short note shows that the transition from strong to weak constraint, accounting for both observations and model biases, reduces the analysis bias.

## 1 Introduction

The perfect model assumption in 4DVar data assimilation, known as strong constraint 4D-Var, is relaxed in weak constraint 4D-Var by adding a forcing term in the model integration to account for model imperfection and to estimate model error. The 4DVar control variable is then augmented by this forcing term and a corresponding term is added to the cost function which constrains model error according to its statistical characteristics. The aim of this short note is to show how the transition from strong to weak constraint affects the quality of the analysis. The demonstration relays on the following 1D advection equation

for a passive tracer describing the evolution of the tracer concentration $q$ on a fixed latitude $\varphi_0$:

$$
\begin{aligned}
\frac{\partial q}{\partial t} + u\frac{\partial q}{\partial \lambda} &= 0, \quad (\lambda, t) \in [0, 2\pi[\times]0, T] \\
\frac{\partial u}{\partial t} &= \frac{\partial u}{\partial \lambda} = 0 \\
q(2\pi, t) &= q(0, t), \quad t \in ]0, T] \\
q(\lambda, 0) &= q_0, \quad \lambda \in [0, 2\pi]
\end{aligned}
\tag{1}
$$

where $(q(\lambda, t), u)^T$ is the state vector representing the tracer concentration $q(\lambda, t)$ at time $t$ and longitude $\lambda$ and the invariant wind $u$.

## 2 Direct, Tangent Linear and Adjoint models

Eq. 1 is discretized using the finite difference formulation with first-order Euler forward time stepping and second order central differencing for the spatial derivatives following Allen et al. (2013). For the sake of simplicity, the time step $\Delta t$ is set equal to the grid spacing $\Delta\lambda$. The model grid is constructed using three longitudes $(\lambda_j, j = 1, 2, 3)$, in which case the discretized tracer concentration is defined as $q_j^n = q(\lambda_j, t_n)$ at time $t_n$, where $n = 0, ..., N-1$ are the time steps of the discretized model with $t_{n+1} = t_n + \Delta t$.





Eq.1 is discretized as follows:

$$\frac{q_j^{n+1} - q_j^n}{\Delta t} + u\frac{q_{j+1}^n - q_{j-1}^n}{2\Delta\lambda} = 0 \quad \text{(with } \Delta t = \Delta\lambda\text{)} \tag{2}$$

Given a discretized state vector $(q_{j-1}^n, q_j^n, q_{j+1}^n)^T$, Eq.2 can then be rewritten as

$$q_{j-1}^{n+1} = q_{j-1}^n - \frac{u}{2}q_j^n + \frac{u}{2}q_{j+1}^n$$

$$q_j^{n+1} = \frac{u}{2}q_{j-1}^n + q_j^n - \frac{u}{2}q_{j+1}^n \tag{3}$$

$$q_{j+1}^{n+1} = -\frac{u}{2}q_{j-1}^n + \frac{u}{2}q_j^n + q_{j+1}^n$$

The discretized model in matrix form $\mathcal{M}$ describing the dynamic evolution from $q_0$ to $q(\Delta t)$ is then given by

$$\mathcal{M}_{0\to\Delta t} = \begin{pmatrix} 1 & -\frac{u}{2} & \frac{u}{2} \\ \frac{u}{2} & 1 & -\frac{u}{2} \\ -\frac{u}{2} & \frac{u}{2} & 1 \end{pmatrix} \tag{4}$$

The equation of the tangent linear model is given by

$$\frac{\partial\hat{q}}{\partial t} + u\frac{\partial\hat{q}}{\partial\lambda} = 0 \tag{5}$$

The discretized equation of the tangent linear model is given by

$$\frac{\hat{q}_j^{n+1} - \hat{q}_j^n}{\Delta t} + u\frac{\hat{q}_{j+1}^n - \hat{q}_{j-1}^n}{2\Delta\lambda} = 0 \tag{6}$$

The discretized tangent linear model in matrix form $\mathbf{M}$ describing the dynamic evolution of the initial perturbation $\hat{q}(0)$ leading to the perturbation $\hat{q}(\Delta t)$ is given by

$$\mathbf{M}_{0\to\Delta t} = \begin{pmatrix} 1 & -\frac{u}{2} & \frac{u}{2} \\ \frac{u}{2} & 1 & -\frac{u}{2} \\ -\frac{u}{2} & \frac{u}{2} & 1 \end{pmatrix} \tag{7}$$

The adjoint model in matrix form is :

$$\mathbf{M}_{\Delta t\to 0}^T = \begin{pmatrix} 1 & \frac{u}{2} & -\frac{u}{2} \\ -\frac{u}{2} & 1 & \frac{u}{2} \\ \frac{u}{2} & -\frac{u}{2} & 1 \end{pmatrix} \tag{8}$$

## 3   Strong constraint 4DVar

Assuming that $Eq.\ 1$ gives an accurate representation of the true system evolution, the aim here is to account for systematic differences between the observations and model by considering the augmented state vector $x(t) = (q(\lambda, t), \beta)^T$, where $\beta$



represents an invariant observation bias. The discretized state is then defined as $x_j^n = (q(\lambda_j, t_n), \beta)^T$ at time $t_n$. The strong-constraint 4D-Var cost function is given by

$$J(x_0) = \frac{1}{2}[x_0 - x_b(0)]^T \mathbf{B}^{-1}[x_0 - x_b(0)] + \frac{1}{2}\sum_{n=0}^{N-1}[\mathcal{H}(q^n) + \beta - q_{obs}^n]^T \mathbf{R}^{-1}[\mathcal{H}(q^n) + \beta - q_{obs}^n]$$

$$= \frac{1}{2}[x_0 - x_b(0)]^T \mathbf{B}^{-1}[x_0 - x_b(0)] + \frac{1}{2}\sum_{n=0}^{N-1}[\tilde{\mathcal{H}}(x^n) - q_{obs}^n]^T \mathbf{R}^{-1}[\tilde{\mathcal{H}}(x^n) - q_{obs}^n]$$

(9)

- $x_0 = (q_0, \beta)^T$ is the control vector.

- $q_{obs}^n$ are the tracer observations where $n = 0, ..., N-1$ are the time steps of the discretized model.

- $\mathcal{H}$ is the observation operator with respect to the state vector $q$ and $\tilde{\mathcal{H}}$ is the observation operator with respect to the augmented state vector $x$.

- $\mathbf{R}$ is the observation error covariance matrix.

- $\mathbf{B}$ is the background error covariance matrix.

- $x^{n+1} = \tilde{\mathcal{M}}_{t_n \to t_{n+1}}(x^n)$, where $\tilde{\mathcal{M}}$ corresponds to the model operator with respect to the augmented state vector $x$.

The evaluation of the solution for tracer observations for all longitudes, is performed under the following idealized setup:

- The tracer observation vector $q_{obs}$ is constructed for all longitudes at the time $t_1 = \Delta t$ as follows

$$q_{obs} = \mathcal{H}(q_t(t_1)) + \beta_t$$

$$= \mathcal{H}(\mathcal{M}(q_t(0))) + \beta_t$$

(10)

where $q_t$ and $\beta_t$ are the true values of the tracer and the observation bias, respectively.

- The background error covariance matrix $\mathbf{R}$ is assumed diagonal. The observation error standard deviations are set equal to $\sigma_{obs}$.

- The background error covariance matrix $\mathbf{B}$ is assumed diagonal. The tracer background error standard deviations are set equal to $\sigma_q$ and the observation bias error standard deviation is set equal to $\sigma_\beta$.

- The background value for the bias $\beta_b$ is assumed equal to zero.

The observation operator with respect to the state vector $q$ is given by

$$\mathcal{H} = \begin{pmatrix} 1 & 0 & 0 \\ 0 & 1 & 0 \\ 0 & 0 & 1 \end{pmatrix}$$

(11)





The observation operator $\tilde{\mathcal{H}}$ with respect to the augmented state vector $x$ is given by

$$\tilde{\mathcal{H}} = \tilde{\mathbf{H}} = \begin{pmatrix} 1 & 0 & 0 & 1 \\ 0 & 1 & 0 & 1 \\ 0 & 0 & 1 & 1 \end{pmatrix} \tag{12}$$

The innovation vector is given by

$$\begin{aligned} d = q_{obs} - \mathcal{H}(q_b(t_1)) &= q_{obs} - \mathcal{H}\mathcal{M}_{0\to\Delta t}(q_b(0)) \\ &= \mathcal{H}\mathcal{M}_{0\to\Delta t}(q_t(0)) + \beta_t - \mathcal{H}\mathcal{M}_{0\to\Delta t}(q_b(0)) \\ &= \mathbf{H}\mathbf{M}_{0\to\Delta t}(q_t(0) - q_b(0)) + \beta_t, \quad \text{where } \mathbf{H} = \mathcal{H} = \begin{pmatrix} 1 & 0 & 0 \\ 0 & 1 & 0 \\ 0 & 0 & 1 \end{pmatrix} \end{aligned} \tag{13}$$

The strong constraint 4D-Var analysis at the initial time $t_0$ is given by

$$\begin{aligned} x_a(0) &= x_b(0) + \delta x_a(0) \\ &= x_b(0) + \frac{\mathbf{B}\tilde{\mathbf{M}}_{\Delta t\to 0}^T \tilde{\mathbf{H}}^T d}{\tilde{\mathbf{H}}\tilde{\mathbf{M}}_{0\to\Delta t}\mathbf{B}\tilde{\mathbf{M}}_{\Delta t\to 0}^T \tilde{\mathbf{H}}^T + \mathbf{R}} \end{aligned} \tag{14}$$

where $\tilde{\mathbf{M}}_{0\to\Delta t} = \tilde{\mathcal{M}}_{0\to\Delta t} = \begin{pmatrix} 1 & -\frac{u}{2} & \frac{u}{2} & 0 \\ \frac{u}{2} & 1 & -\frac{u}{2} & 0 \\ -\frac{u}{2} & \frac{u}{2} & 1 & 0 \\ 0 & 0 & 0 & 1 \end{pmatrix}$

The observation bias $\beta$ is invariant $(\tilde{\mathbf{M}}_{0\to\Delta t}(\beta) = \tilde{\mathcal{M}}_{0\to\Delta t}(\beta) = \beta)$. Note that $\beta$ is constant in space and time but its value can be incremented in the assimilation process.

The analysis at the time $t_1$ is given by

$$\begin{aligned} x_a(t_1) &= \tilde{\mathcal{M}}_{0\to\Delta t}(x_b(0)) + \tilde{\mathbf{M}}_{0\to\Delta t}\delta x_a(0) \\ &= x_b(t_1) + \frac{\tilde{\mathbf{M}}_{0\to\Delta t}\mathbf{B}\tilde{\mathbf{M}}_{\Delta t\to 0}^T \tilde{\mathbf{H}}^T d}{\tilde{\mathbf{H}}\tilde{\mathbf{M}}_{0\to\Delta t}\mathbf{B}\tilde{\mathbf{M}}_{\Delta t\to 0}^T \tilde{\mathbf{H}}^T + \mathbf{R}} \end{aligned} \tag{15}$$

## 4 Weak constraint 4DVar

In contrast to strong constraint 4DVar, weak constraint accounts for the deviation of $Eq.$ 1 from a true representation of the system evolution. The tracer observation vector $q_{obs}$ for all longitudes at the time $t_1 = \Delta t$ is, instead, constructed using the following equation:

$$\begin{aligned} q_{obs} &= \mathcal{H}(q_t(t_1)) + \beta_t \\ &= \mathcal{H}(\mathcal{M}_t(q_t(0))) + \beta_t \end{aligned} \tag{16}$$





where $\mathcal{M}_t$ is the discretized true model of the following equation:

$$\frac{\partial q}{\partial t} + u\frac{\partial q}{\partial \lambda} = k\frac{\partial^2 q}{\partial \lambda^2}, \quad (\lambda, t) \in [0, 2\pi[\times]0, T]$$

$$q(2\pi, t) = q(0, t), \quad t \in ]0, T] \tag{17}$$

$$q_0 = q_t(0)$$

where $k$ is the diffusion coefficient.

Eq. 17 is discretized using the finite difference formulation with first-order Euler forward time stepping and second order central differencing for the spatial derivatives with $\Delta t = \Delta \lambda$.

The discretized true model is then given by

$$\mathcal{M}_t = \begin{pmatrix} 1 - 2\kappa & -\frac{u}{2} + \kappa & \frac{u}{2} + \kappa \\ \frac{u}{2} + \kappa & 1 - 2\kappa & -\frac{u}{2} + \kappa \\ -\frac{u}{2} + \kappa & \frac{u}{2} + \kappa & 1 - 2\kappa \end{pmatrix} \quad \text{where } \kappa = \frac{k\Delta t}{(\Delta \lambda)^2} = \frac{k}{\Delta \lambda} \quad (\Delta \lambda = \frac{2\pi}{3}) \tag{18}$$

The weak constraint 4D-Var cost function is given by

$$J(x_0, x(t_1)) = \frac{1}{2}[x_0 - x_b(0)]^T \mathbf{B}^{-1}[x_0 - x_b(0)] + \frac{1}{2}[\tilde{\mathcal{H}}(x(t_1)) - q_{obs}]^T \mathbf{R}^{-1}[\tilde{\mathcal{H}}(x(t_1)) - q_{obs}]$$
$$+ \frac{1}{2}\eta^T \mathbf{Q}^{-1}\eta \tag{19}$$

where $(x_0, x(t_1))^T = (q_0, q(t_1), \beta)^T$ is the control vector, $\eta = q(t_1) - \mathcal{M}_{0 \to \Delta t}(q_0)$ is the tracer model error, and $\mathbf{Q}$ is the tracer model error covariance matrix, which is assumed diagonal with model error standard deviations equal to $\sigma_m$.

For the sake of simplicity, the background value of the tracer model error $\eta_b$ is assumed equal to zero (i.e., $q_b(t_1) = \mathcal{M}_{0 \to \Delta t}(q_b(0))$).

The background error covariance matrix and the tracer model error covariance matrix are given by

$$\mathbf{B} = \begin{pmatrix} \sigma_q^2 & 0 & 0 & 0 \\ 0 & \sigma_q^2 & 0 & 0 \\ 0 & 0 & \sigma_q^2 & 0 \\ 0 & 0 & 0 & \sigma_\beta^2 \end{pmatrix} \quad \text{and} \quad \mathbf{Q} = \begin{pmatrix} \sigma_m^2 & 0 & 0 \\ 0 & \sigma_m^2 & 0 \\ 0 & 0 & \sigma_m^2 \end{pmatrix} \tag{20}$$

The value of the weak constraint 4D-Var analysis at $t_1 = \Delta t$ is then given by

$$x_a(t_1) = x_b(t_1) + \frac{(\tilde{\mathbf{M}}_{0 \to \Delta t}\mathbf{B}\tilde{\mathbf{M}}_{\Delta t \to 0}^T + \tilde{\mathbf{Q}})\tilde{\mathbf{H}}^T d}{\tilde{\mathbf{H}}(\tilde{\mathbf{M}}_{0 \to \Delta t}\mathbf{B}\tilde{\mathbf{M}}_{\Delta t \to 0}^T + \tilde{\mathbf{Q}})\tilde{\mathbf{H}}^T + \mathbf{R}} \tag{21}$$

where $d = \mathcal{H}\mathcal{M}_t(q_t(0)) + \beta_t - \mathcal{H}\mathcal{M}_{0 \to \Delta t}(q_b(0))$ and $\tilde{\mathbf{Q}} = \begin{pmatrix} \mathbf{Q} & 0 \\ 0 & 0 \end{pmatrix}$

The computation of the analysis is performed relaying on the following setup: $u = 1$, $k = 0.4$, $\beta_t = 0.2$, $q_b(0) = (1, 2, 3)^T$, $q_t(0) = (1.1, 2.2, 3.3)^T$, $\sigma_{obs} = 0.01$ and $\sigma_m = \sigma_q = \sigma_\beta = 0.1$.





The strong and weak constraint analyses for the tracer at $t_1$ are $(1.5, 1, 3.7)^T$ and $(2.2, 1, 3.2)^T$, respectively.

The analysis accuracy is measured by the normalized analysis error (NAE), which is defined as the mean absolute difference

between the analysis and the truth divided by the truth. A NAE of 0% indicates a perfect match of the analysis with the truth.

$$NAE = \frac{1}{3}\sum_{j=1}^{3}\frac{|q_a(\lambda_j, t_1) - q_t(\lambda_j, t_1)|}{q_t(\lambda_j, t_1)} \qquad (22)$$

The weak constraint outperforms the strong constraint, with NAE of 1.7% compared with 4.4%.

## 5   Summary

4DVar assumes random zero-mean random errors. Therefore, to build an unbiased analysis, variational bias correction and

weak constraint are designed to conjointly remove observations and model biases as shown in Fig.1. Throughout an idealised

experiment, this note demonstrates the added-value of weak constraint in the reduction of the analysis bias when compared to

the strong constraint.

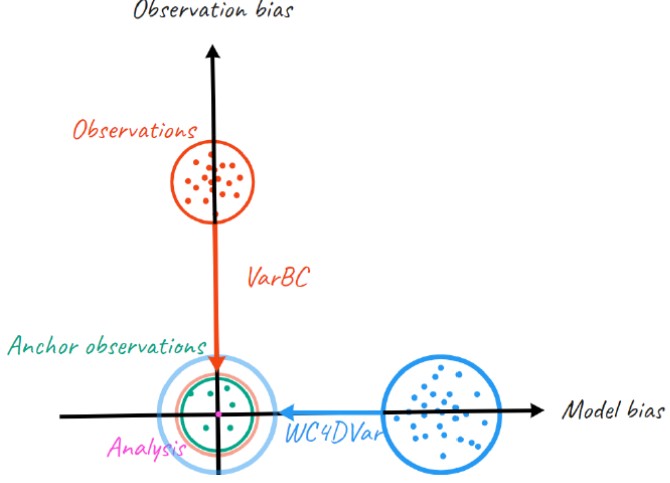

**Figure 1.** Unbiased observations are referred to as *anchor observations*. Variational bias correction (*VarBC*) and weak constraint 4DVar (*WC4DVar*) conjointly target an unbiased analysis. Courtesy of Patrick Laloyaux, ECMWF.

*Data availability.* Data sharing is not applicable to this article as no new data were created or analyzed in this study.



*Author contributions.* The author confirms sole responsibility for the following: study conception and design, analysis and interpretation of
105 results, and manuscript preparation.

*Competing interests.* The contact author declares no competing interests.



# References

Allen, D. R., Hoppel, K. W., Nedoluha, G. E., Kuhl, D. D., Baker, N. L., Xu, L., and Rosmond, T. E.: Limitations of wind extraction from 4D-Var assimilation of ozone, Atmospheric Chemistry and Physics, 13, 3501–3515, https://doi.org/10.5194/acp-13-3501-2013, 2013.