# Peer review of "On the transition from strong to weak constraint 4DVar using a simple one-dimensional advection equation for a passive tracer"

_EGUsphere, 2023_

## Referee Comment (RC2)

Review of "On the transition from strong to weak constraint 4DVar using a simple one-dimensional advection equation for a passive tracer," by Noureddine Semane

**General Comments**

This short article compares strong and weak constraint 4DVar in the context of a simple 1-D advection equation for a passive tracer on a 3-point grid. The simplified forward model allows an analytic solution to both approaches. The results illustrate with a single example that the analysis error is reduced when accounting for errors in both the model and the observations. The results are interesting and useful. However, there is only a single case shown. It would be helpful to provide some more case studies to show the robustness of the results. In addition, the notation is sometimes confusing and could use some modifications. Finally, the discussion of Figure 1 is very sparse, and this figure needs to be explained in more detail if it is to be included.

**Specific Comments**

Eq. 1: You use $q_0$ where the time index is in the subscript, right? But later you have the time index in the superscript. To be consistent, maybe use $q^0$ here. The time goes from 0 to $T$ here, but $T$ is not defined. What does the "[x]" symbol mean in the first line of Eq. 1? Maybe it would be easier to separate the two ranges.

Line 12: Since u is constant in space and not allowed to change in the 4D-Var, maybe it is better to leave it out of the state vector here and just consider it a constant. Indeed, on line 22 you exclude u from the state vector without any explanation. Alternately, as mentioned in comment RC1, you could actually include u in your state vector and allow it to be modified.

Line 16: Does the system also work if $\Delta t \neq \Delta \lambda$?

Line 16-19: May want to move the sentence, "For the sake of simplicity…" to the end of this paragraph and explicitly say what the grid spacing is, $\Delta \lambda = 2\pi/3$. You say that later, but would be good to have here.

Line 18: You are only actually performing one time step, right? So N=2. It might be helpful to say that here and address the times as n=0 and n=1 throughout.

Line 19: It is confusing that sometimes you use times as 0 and $\Delta t$, while other times as $t_0$ and $t_1$. Maybe to be clear it would be helpful to say explicitly that $t_0 = 0$ and $t_1 = \Delta t$.

Line 24: I think you need to clarify your notation here. I think you want $q_0$ and $q(\Delta t)$ to be vectors of length 3, right, as specified in line 22. In the GMD submission instructions (https://www.geoscientific-model-development.net/submission.html#math) it says, "Matrices are printed in boldface, and vectors in boldface italics." So maybe on line 22 you should write:

$$\boldsymbol{q}^n = \left(q_{j-1}^n, q_j^n, q_{j+1}^n\right)^T$$

Then use notation $\boldsymbol{q}^0$ and $\boldsymbol{q}^1$ on line 24.

Line 25: If you make the above changes, then you could have the model go from time index 0 to 1 rather than 0 to $\Delta t$.

Line 26: You may want to point out here that since this is a linear model the TLM is exactly the same as the full model.

Line 30: Does the carat (^) represent the perturbation from a background forecast? Here you could write the vectors as $\hat{\boldsymbol{q}}^0$ and $\hat{\boldsymbol{q}}^1$.

Line 36: You say that Eq. 1 represents the true system evolution, but isn't the true evolution actually given by the discretized version of Eq. 1, i.e., by Eq. 2?

Line 38: Since x is a vector, I would suggest to also make this explicit as:

$$\boldsymbol{x}^n = (\boldsymbol{q}^n, \beta)^T = \left(q_{j-1}^n, q_j^n, q_{j+1}^n, \beta\right)^T$$

Line 38: You say that $\beta$ represents an invariant observation bias. What exactly do you mean by invariant? Oh, I see that you explain this later on lines 65-66. May want to say that here.

Eq. 9: The notation is again confusing, since $x_0$ has the time index in the subscript, but in line 38 you have the time index in the superscript. Then in $x_b(0)$ the time index is in parentheses, and there is no indication that this includes three grid points. Would making these vectors symbols bold and have the time index in the superscript make sense?

Are there observations at both time 0 and time Δt? Later (line 49) it looks like observations are only taken at time $t_1 = \Delta t$, right? So although the summation is used in Eq. 9, observations are only for n=1.

Line 47: Might be helpful write out the model operator with respect to the augmented state vector here. I see that you put it later in Eq. 14, but I think it would be helpful for the reader to see this earlier.

Line 51: q refers to all three points here, right? So why not use vector notation (bold, non-italics)? Also, should have the time index on $\boldsymbol{q}_{obs}^1$ to be consistent with notation in Eq. 9.

Line 57: Here again I would suggest using vector notation for q.

Eq. 13: The notation is again confusing here. I think using bold for vectors and having the time index in the superscript would help. This comment also applies to all future equations.

Eq. 17: So does this model now represent the truth rather than Eq. 1?

Line 83: Why do you write the control vector as having both times 0 and 1 here? Before you had it with only time 0. Isn't the state at time 1 just the forward integration from time 0 using Eq. 18?

Line 93: Could you also write out the values of $q_t(t_1)$?

A single example isn't very satisfying. I understand this is a short note, and a complete sensitivity analysis may be beyond the scope. But is it obvious that error reduction occurs for a wide range of initial conditions and values of the wind and diffusion coefficients? Are there possible conditions where weak constraint would do worse than strong constraint?

Figure 1: This figure requires a lot more explanation if it is to be used in the paper. What do all the dots refer to? I think the red dots and blue dots are observations, but what are the green dots? What are "anchor observations", and how would you know that any given observation was unbiased? What do the cyan and orange circles represent? Is the analysis the single point in the middle?

**Technical Corrections**

Line 2: "Relaying" should be "Relying"

Line 9: "relays" should be "relies"

Eq. 1: Is the left bracket reversed on line 3?

Line 36: Here "*Eq*" is in italics, but other places it is non-italics.

Eq. 17: Is the left bracket reversed on line 2?

Line 92: "relaying" should be "relying"

Line 99: Probably want to remove one occurrence of "random" in this sentence.